# A Single-Component Multilayered Self-Assembling Protein Nanoparticle Vaccine Based on Extracellular Domains of Matrix Protein 2 against Both Influenza A and B

**DOI:** 10.3390/vaccines12090975

**Published:** 2024-08-28

**Authors:** Yi-Nan Zhang, Keegan Braz Gomes, Yi-Zong Lee, Garrett Ward, Bomin Xie, Sarah Auclair, Linling He, Jiang Zhu

**Affiliations:** 1Department of Integrative Structural and Computational Biology, The Scripps Research Institute, La Jolla, CA 92037, USA; yizhang@scripps.edu (Y.-N.Z.); ylee@scripps.edu (Y.-Z.L.); gaward@scripps.edu (G.W.); bxie@scripps.edu (B.X.); sauclair@scripps.edu (S.A.); linling@scripps.edu (L.H.); 2Uvax Bio, LLC, Newark, DE 19702, USA; keegan.brazgomes@uvaxbio.com; 3Department of Immunology and Microbiology, The Scripps Research Institute, La Jolla, CA 92037, USA

**Keywords:** antibody, influenza A, influenza B, M2e, protein nanoparticle, universal influenza vaccine

## Abstract

The development of an effective and broadly protective influenza vaccine against circulating and emerging strains remains elusive. In this study, we evaluated a potentially universal influenza vaccine based on single-component self-assembling protein nanoparticles (1c-SApNPs) presenting the conserved matrix protein 2 ectodomain (M2e) from influenza A and B viruses (IAV and IBV, respectively). We previously designed a tandem antigen comprising three IAV M2e domains of human, avian/swine, and human/swine origins (termed M2ex3). The M2ex3-presenting 1c-SApNPs conferred complete protection in mice against sequential lethal challenges with H1N1 and H3N2. To broaden this protection to cover IBVs, we designed a series of antigens incorporating different arrangements of three IAV M2e domains and three copies of IBV M2e. Tandem repeats of IAV and IBV (termed influenza A-B) M2e arrayed on the I3-01v9a 60-mer 1c-SApNP, when formulated with an oil-in-water emulsion adjuvant, generated greater M2e-specific immunogenicity and protective efficacy than the soluble influenza A-B M2e trimer, indicated by higher survival rates and reduced weight loss post-challenge. Importantly, one of the influenza A-B M2e SApNP constructs elicited 100% protection against a lethal influenza A/Puerto Rico/8/1934 (H1N1) challenge in mice and 70% protection against a lethal influenza B/Florida/4/2006 (Yamagata lineage) challenge, the latter of which has not been reported in the literature to date. Our study thus provides a promising M2e-based single-component universal vaccine candidate against the two major types of influenza virus circulating in humans.

## 1. Introduction

Influenza is an acute respiratory illness caused by influenza viruses belonging to the Orthomyxoviridae family [1,2,3,4,5], with 3–5 million severe disease cases and 290,000–650,000 deaths worldwide each year [6]. The influenza virus is an enveloped negative-sense single-stranded RNA virus with a segmented genome that can be classified as group A, B, C, or D, with influenza A viruses (IAVs) and influenza B viruses (IBVs) causing most infections in humans. IAVs can be further classified into H1-18 and N1-11 subtypes based on the antigenic properties of the two surface glycoproteins hemagglutinin (HA) and neuraminidase (NA), respectively [4]. IBVs have a single HA and NA subtype but are classified into two lineages, Victoria and Yamagata [1,2]. IAVs can infect a plethora of hosts, whereas IBVs are restricted to humans [7]. Both IAVs and IBVs can evade prior immunity and cause seasonal epidemics through antigenic drift, an evolutionary process in which small changes are introduced into HA and NA under immune pressure. IAVs account for approximately 75% of all human infections and can cause global pandemics through antigenic shift, in which HAs and NAs from different host species recombine to form novel strains against which the general population lacks preexisting immunity [8,9].

To combat influenza, current seasonal flu vaccines are quadrivalent and include two IAV subtypes, H1N1 and H3N2, and two IBVs, one each from the Victoria and Yamagata lineages [10,11]. However, these vaccines are strain-specific and must be updated yearly to include predicted strains [8]. As a result, strain mismatch, along with mutations acquired during vaccine production in eggs, causes substantial variation in vaccine efficacy (10–60%), highlighting the urgent need for better vaccines capable of broader protection against diverse circulating and emerging influenza viruses [12,13]. A potential target for universal influenza vaccine development is the extracellular domain of the highly conserved matrix protein 2 ectodomain (M2e), a proton-selective viroporin [14,15,16,17,18,19]. However, M2e is a small peptide (~23 amino acids [aa] for IAV and ~8 aa for IBV) and is significantly less immunogenic than the larger HA and NA surface glycoproteins. To improve the immunogenicity of M2e, it can be attached to large carriers, as previously reported for influenza A M2e-based vaccines. These vaccines elicit M2e-specific antibodies that can cross-react against different IAV subtypes, thereby reducing virus replication and egress [14,15,16,17,18,20,21,22]. Unlike HA and NA, which elicit neutralizing antibodies, anti-M2e antibodies are non-neutralizing and confer protection via FcγR-dependent mechanisms, such as antibody-dependent cellular cytotoxicity and antibody-dependent cellular phagocytosis [18,23,24,25]. Multiple influenza A M2e vaccine candidates have been evaluated in phase 1 human trials, but low immunogenicity and durability have hindered further development [16,26].

We recently developed a tandem antigen consisting of IAV M2e of human, avian/swine, and human/swine origins (M2ex3) and incorporated it into our single-component self-assembling protein nanoparticle (1c-SApNP) platform [27]. This platform approach was largely inspired by the success of virus-like particle (VLP) vaccines, which are typically made of self-assembling viral proteins used as vaccines against cognate viruses or as a carrier to improve the immunogenicity of subdominant antigens [28,29,30,31,32,33,34,35]. Protein nanoparticles can be designed to mimic the desirable size and antigen display characteristics of VLP vaccines, with the added advantage of greater design flexibility, allowing them to be fine-tuned to target diverse pathogens. Our SApNPs, which are produced entirely from a single plasmid, can be expressed in Chinese hamster ovary (CHO) cells in a single step with high yield and purity and have been successfully applied to human immunodeficiency virus 1 [36,37,38], hepatitis C virus [39], influenza [27], Ebola virus [40], and severe acute respiratory syndrome coronavirus 2 (SARS-CoV-2) [41,42] vaccine development. Results from our previous study showed that SApNPs displaying IAV M2ex3 and adjuvanted with AddaVax induced robust M2e-specific antibody responses [27]. These M2ex3 SApNP vaccines conferred complete protection against consecutive heterologous challenges with lethal doses of influenza A H1N1 and H3N2 in mice [27]. Notably, these M2ex3 SApNPs were retained in lymph nodes for 8 weeks or longer, inducing long-lasting germinal center reactions and M2e-specific functional T cell responses.

IBVs, although accounting for only ~25% of seasonal influenza cases, are responsible for 52% of influenza-related deaths in children 18 years of age and younger [43,44]. Therefore, IBVs must also be considered in the development of a truly “universal” influenza vaccine, which has remained a challenge for decades. Several groups have developed vaccine candidates combining M2e sequences from multiple IAV species [16,20,21,45,46,47,48,49], but there are no previously reported vaccine constructs that have incorporated both IAV and IBV M2e to target both influenza A and B, likely in part due to the small size of IBV M2e (~8 aa), which is believed to be insufficient to mount an antibody response.

Here, we demonstrated that displaying tandem repeats of IAV M2e and IBV M2e on 1c-SApNPs elicited up to 100% protection against a lethal influenza A H1N1 challenge in mice and, more significantly, up to 70% protection against a lethal influenza B challenge, which is the first time IBV M2e-mediated protection has been demonstrated. In summary, our influenza A-B M2e SApNP vaccines induced strong IAV and IBV M2e-specific immunogenicity and protected against both influenza A and B challenges in mice, demonstrating the potential to be a truly universal M2e-based single-component influenza vaccine candidate and warranting further development.

## 2. Materials and Methods 

### 2.1. Expression and Purification of Various Influenza A-B M2e Immunogens

Rationally designed influenza A-B M2e immunogens were produced to facilitate in vitro and in vivo characterization. Influenza A-B M2e-scaffolded trimers and SApNPs were transiently expressed in ExpiCHO cells (Thermo Fisher, Waltham, MA, USA) using a previously described protocol [27]. Briefly, ExpiCHO cells were thawed and incubated with ExpiCHO Expression Medium (Thermo Fisher) in a shaker incubator at 37 °C at 135 rotations per minute (rpm) with 8% CO_2_. When cells reached a density of 10 × 10^6^ cells/mL, ExpiCHO Expression Medium was used to dilute the cell density to 6 × 10^6^ cells/mL for transfection. An ExpiFectamine CHO/plasmid DNA mixture was prepared for 100 mL transfections in ExpiCHO cells according to the manufacturer’s instructions. For the trimer and four SApNP constructs evaluated in this study, 100 μg of antigen plasmid and 320 μL of ExpiFectamine CHO reagent were mixed in 7.7 mL of cold OptiPRO medium (Thermo Fisher). After the first feed on day 1, ExpiCHO cells were cultured in a shaker incubator at 32 °C at 120 rpm with 8% CO_2_ following the Max Titer protocol with an additional feed on day 5 (Thermo Fisher). Culture supernatants were harvested 13–14 days post-transfection, clarified by centrifugation at 4000 rpm for 20 min, and filtered using a 0.45 μm filter (Thermo Fisher). Expressed M2e immunogens were extracted from the culture supernatants using an IAV M2e-specific mAb148 antibody column. The bound protein was eluted three times by 5 mL of glycine buffer (0.2 M glycine, pH 2.2) and neutralized by adding 0.375 mL of Tris-base buffer (2 M Tris, pH 9.0). Eluates were pooled and buffer-exchanged via ultracentrifugal filtration to phosphate-buffered saline (PBS). The sizes of the trimers and SApNPs were analyzed by size exclusion chromatography (SEC) using ÄKTA pure 25 (Cytiva, Marlborough, MA, USA). The trimer was further purified by a Superdex 75 Increase 10/300 GL column (Cytiva, catalog no. 29148721), and SApNPs were characterized on a Superose 6 Increase 10/300 GL column (Cytiva, catalog no. 29091596). Protein concentration was determined based on ultraviolet absorbance at 280 nm (UV_280_) with theoretical extinction coefficients. Proteins were frozen in liquid nitrogen and stored at −80 °C until further use.

### 2.2. Sodium Dodecyl Sulfate-Polyacrylamide Gel Electrophoresis (SDS-PAGE) and Blue Native-Polyacrylamide Gel Electrophoresis (BN-PAGE)

The M2e trimer and four SApNPs were analyzed by SDS-PAGE and BN-PAGE. The proteins were mixed with loading dye and added to either a 10% Tris-Glycine Gel (Bio-Rad, catalog no. 561096, Hercules, CA, USA) or 4–12% Bis-Tris NativePAGE gel (Invitrogen, catalog no. BN1003BOX, Carlsbad, CA, USA). For SDS-PAGE under reducing conditions, the proteins were first treated with 6× SDS reducing dye (Thermo Scientific, catalog no. J61337.AD, Waltham, MA, USA) and boiled for 5 min at 100 °C. SDS-PAGE gels were loaded with 2 μg of the sample, and BN-PAGE gels were loaded with 4 μg of the sample. SDS-PAGE gels were run for 50 min at 150 V using SDS running buffer (Bio-Rad, catalog no. 1610732) and BN-PAGE gels were run for 2–2.5 h at 150 V using NativePAGE running buffer (Invitrogen, catalog no. BN2007) according to the manufacturer’s instructions. SDS-PAGE gels were stained using InstantBlue (Abcam, catalog no. ab119211, Cambridge, UK) and BN-PAGE gels were destained using a solution of 6% ethanol and 3% glacial acetic acid. Gel images were taken by Chemi Doc XR^+^ (Bio-Rad) with the Image Lab program (version 6.1.0).

### 2.3. Dynamic Light Scattering (DLS)

The particle size distributions of influenza A-B M2e I3-01v9a-L7P SApNPs were obtained using a Zetasizer Ultra instrument (Malvern, Malvern, Worcestershire, UK). The mAb148-purified SApNPs from ExpiCHO cells were first diluted to 0.2 mg/mL in 1× PBS buffer, and 30 μL of the prepared sample was added to a quartz batch cuvette (Malvern). Particle size was measured at 25 °C using a backscattering mode. Data processing was performed on a Zetasizer Ultra instrument, and the particle size distribution was plotted using GraphPad Prism 10.2.3 software.

### 2.4. Negative-Stain Electron Microscopy (nsEM) Analysis

The nsEM analysis of various influenza A-B M2e I3-01v9a SApNP samples was performed by the Core Microscopy Facility at The Scripps Research Institute. All SApNP samples were prepared at a concentration of 0.005–0.02 mg/mL. Carbon-coated copper grids (400 mesh) were glow-discharged, and 8 μL of each sample was adsorbed for 2 min. The excess sample was wicked away, and grids were stained with 2% uranyl formate for 2 min. Excess stain was wicked away, and grids were allowed to dry. Samples were analyzed at 120 kV with a Talos L120C transmission electron microscope (Thermo Fisher), and images were acquired with a CETA 16 M CMOS camera. All samples purified by mAb148 were imaged under 52,000× magnification before further use.

### 2.5. Enzyme-Linked Immunosorbent Assay (ELISA) for Antibody Binding

Ninety-six-well high-binding, flat-bottom, half-area plates (Corning, Corning, NY, USA) were first coated with 50 µL/well of 0.1 μg of the M2e trimer or SApNP antigens. The plates were then incubated at 4 °C overnight. The next day, the plates were washed five times with PBST wash buffer (1× PBS + 0.05% [*v*/*v*] Tween 20, Promega, Madison, WI, USA). Each well was then blocked with 150 µL of blocking buffer consisting of 1× PBS + 4% (*w*/*v*) nonfat milk (Bio-Rad, catalog no. 1706404XTU). The plates were blocked for 1 h at room temperature and then washed five times with PBST. For antigen binding, antibodies were diluted in blocking buffer to a starting concentration of 10 μg/mL followed by a 10-fold dilution series until 1 pg/mL. Next, 50 μL of each antibody dilution was added to the sample wells. The plates were incubated for 1 h at room temperature and then washed with PBST. Next, 50 µL of horseradish peroxidase-conjugated anti-human secondary antibody (Jackson ImmunoResearch Laboratories, catalog no. 109-035-088, West Grove, PA, USA) at a 1:5000 dilution in PBST was then added to each well. The plates were then incubated with the secondary antibody for 1 h at room temperature and then washed six times. Lastly, the wells were developed with 50 μL of 3,3′,5,5′-tetramethylbenzidine (TMB; Thermo Scientific, catalog no. 34028) for 3 min before the reaction was stopped with 50 μL of 2.0 N sulfuric acid (Aqua Solutions, catalog no. 9100-500ML, Deer Park, TX, USA). The plates were then immediately read on a plate reader (PerkinElmer, Waltham, MA, USA) at a wavelength of 450 nm. Half-maximal effective concentration (EC_50_) values were then calculated from full curves using GraphPad Prism 10.2.3 software.

### 2.6. Propagation of Influenza Viruses

For influenza challenge studies in mice, mouse-adapted A/Puerto Rico/8/1934 (H1N1) virus was obtained from BEI Resources (catalog no. NR-28652; National Institute of Allergy and Infectious Diseases [NIAID], National Institutes of Health [NIH]), and mouse-adapted B/Florida/4/2006 (Yamagata-like) was kindly provided by Dr. Eric A. Weaver [50]. Two human influenza viruses used as strain-matched positive controls for mouse challenge studies were obtained from BEI Resources (NIAID, NIH; IAV, A/Puerto Rico/8/1934 [H1N1], NR-348, and IBV, B/Florida/4/2006 [Yamagata lineage], NR-41795). For the propagation of influenza viruses, 800,000 cells/well of Madin-Darby canine kidney (MDCK) cells (CCL-34; American Type Culture Collection) were plated in six-well cell culture-treated plates at 37 °C overnight. The next day, MDCK cells were infected with the stock virus in serum-free Dulbecco’s Modified Eagle Medium (DMEM; Thermo Fisher, catalog no. 10313021) containing 1% penicillin/streptomycin and 1% L-glutamine at 37 °C for 1 h. The inoculum was then removed, and the plates were washed with PBS. Next, 2 mL of DMEM infection media containing 0.2% (*w*/*v*) bovine serum albumin (BSA; Sigma Aldrich, catalog no. A7906-50G) and 1 μg/mL L-1-tosylamido-2-phenylethyl chloromethyl ketone (TPCK)-treated trypsin (Sigma Aldrich, catalog no. 4370285) was added to each well, and plates were incubated at 37 °C for 65–72 h. After incubation, the supernatant from each well was collected, centrifuged at 4000 rpm for 10 min, aliquoted, and stored at −80 °C. The viruses were then passaged two more times and stored at −80 °C until use. A standard hemagglutination assay was performed to determine the hemagglutination titers of the propagated viruses.

### 2.7. Influenza Virus Inactivation

To prepare the positive controls of strain-matched mouse-adapted influenza viruses for challenge studies in mice, both A/Puerto Rico/8/1934 (H1N1) and B/Florida/4/2006 (Yamagata lineage) were propagated in six-well tissue culture-treated plates as described above. The supernatant was collected and filtered through a 0.45 μm filter. Viruses were then suspended in a 0.02% *w*/*v* formaldehyde solution (Sigma Aldrich, catalog no. 252549-500ML, Burlington, MA, USA) with 10% (*w*/*v*) sucrose (EMD Millipore) in PBS and gently shaken at 4 °C for 5 days for inactivation. Next, virus supernatants were gently added onto a 30% (*w*/*v*) sucrose cushion and ultracentrifuged at 30,000 rpm at 4 °C for 1.5 h. The supernatant was disposed of, and the virus pellet was resuspended in PBS, aliquoted, and stored at −80 °C until the immunization studies. The concentration of the inactivated virus was quantified using a bicinchoninic acid assay (Thermo Scientific, catalog no. 23225), and the hemagglutination of inactivated virus was confirmed using a hemagglutination assay. Virus inactivation was confirmed using a 50% tissue culture infection dose (TCID_50_) assay.

### 2.8. Mouse Immunizations, Sample Collection, and Viral Challenge

Six- to eight-week-old female BALB/c mice were purchased from The Jackson Laboratory and housed in ventilated cages in environmentally controlled rooms in the Immunology building at The Scripps Research Institute (TSRI). All animal studies followed the Association for the Assessment and Accreditation of Laboratory Animal Care (AAALAC) guidelines and used an approved Institutional Animal Care and Use Committee (IACUC) protocol. Mice were immunized intradermally through footpad injections at weeks 0 and 3 with 80 μL of antigen/adjuvant mix containing 10 μg of influenza A-B M2e-based vaccine antigens in 40 μL PBS and 40 μL of AddaVax adjuvant (InvivoGen, catalog no. vac-adx-10, San Diego, CA, USA). Mice were immunized intradermally through four footpads with 20 μL of antigen/adjuvant per footpad using a 29-gauge insulin needle under 3% isoflurane anesthesia with oxygen. For positive control groups, mice received 10 μg of either inactivated A/Puerto Rico/8/1934 or B/Florida/4/2006. Blood was collected at weeks 2 and 5. To isolate serum from whole blood, blood was allowed to clot and then centrifuged at 14,000 rpm for 10 min to isolate the serum. The serum was then heat-inactivated at 56 °C for 30 min and centrifuged at 8000 rpm for 10 min. The supernatant was collected and evaluated for M2e-specific antibody binding.

Benchmark challenge studies were performed for mouse-adapted A/Puerto Rico/8/1934 (H1N1) and mouse-adapted B/Florida/4/2006 (Yamagata lineage) to determine the lethal dose 50 (LD_50_) in mice. First, various dilutions of the virus were administered to mice intranasally (25 μL per nostril). Survival, weight loss, and morbidity were monitored for 14 days post-challenge. Mice with weight loss > 25% of their total body weight were euthanized. Reed–Muench and Spearman–Karber methods [51,52] were utilized to calculate LD_50_ values. To assess the protective efficacy conferred by influenza A-B M2e-based immunogens co-formulated with AddaVax using prime-boost immunization, mice were challenged with the LD_50_ × 10 of mouse-adapted A/Puerto Rico/8/1934 (H1N1) or LD_50_ × 5 of mouse-adapted B/Florida/4/2006 (Yamagata lineage) at weeks 6 or 10. The two challenge regimens tested included the influenza A H1N1 challenge followed by an influenza B challenge or vice versa. Survival and weight loss were monitored for 14 days post-challenge.

### 2.9. ELISA for Serum Samples

To assess the IAV and IBV M2e-specific binding of vaccine immune-sera, 50 μL of each of M2exA3B3-5GS-foldon, M2exB3A3-5GS-foldon, M2e(AB)x3-5GS-foldon, or M2e(BA)x3-5GS-foldon trimer were coated on 96-well plates at a concentration of 0.1 μg/well and incubated at 4 °C overnight. The plates were washed five times with PBST and blocked with 150 μL of 4% (*w*/*v*) nonfat milk blocking buffer at room temperature for 1 h. The plates were then washed five times, after which 50 μL of M2e immune-sera elicited by trimer or I3-01v9 SApNP constructs were added to each well starting at a 40-times dilution followed by seven 10-fold dilutions. The plates were then incubated at room temperature for 1 h and then washed with PBST. Next, 50 μL of the diluted (1:3000) horseradish peroxidase-conjugated goat anti-mouse immunoglobulin G antibody (Jackson ImmunoResearch Laboratories) in PBST was added to each well. The plates were incubated at room temperature for 1 h and washed six times with PBST. Lastly, 50 μL of TMB substrate was added to each well and incubated for 3 min, after which 50 μL of 2.0 N sulfuric acid (Aqua Solutions) was added to each well. The plates were then read at a wavelength of 450 nm using a plate reader (PerkinElmer), from which EC_50_ values were calculated using GraphPad Prism 10.2.3 software.

### 2.10. Cell-Based ELISA

Cell-based ELISA was used to evaluate the binding of M2e-immune sera to homotetrameric M2e on influenza-infected cells. The following four reagents were obtained from BEI Resources (NIAID, NIH): (1) IAV, A/Puerto Rico/8/1934 (H1N1; NR-348), (2) IAV, A/Hong Kong/1/1968 (H3N2; mother clone; NR-28620), (3) IBV, B/Florida/4/2006 (Yamagata lineage; NR-41795), and (4) IBV, B/Brisbane/60/2008 (Victoria lineage; NR-42005). The viruses, which served as representative strains of influenza A (H1N1 and H3N2 subtypes) and influenza B (Yamagata and Victoria lineages), were grown in MDCK cells using the method previously described for propagating and quantifying challenge strains. For the cell-based ELISA, MDCK cells were plated overnight in 96-well cell culture plates at a density of 1.8 × 10^4^ cells/well. The next day, the cells were washed and infected with 100 μL of one of the four viruses at a multiplicity of infection (MOI) of 0.1. Twenty hours later, the supernatants were removed, and cells were fixed with 100 μL of 3.7% (*w*/*v*) formaldehyde. The cells were then washed, and the ELISA protocol mentioned above was used except for an incubation step with TMB for 7 min.

### 2.11. Statistical Analysis

Data were collected from 10 mice per group in the vaccine and live virus challenge studies and serum binding experiments. Statistical analyses at various time points were performed using one-way analysis of variance (ANOVA) followed by Tukey’s multiple-comparison post hoc test. Statistical significance in the figures is shown as the following: ns (not significant), * *p* < 0.05, ** *p* < 0.01, *** *p* < 0.001, and **** *p* < 0.0001. The figures were generated using GraphPad Prism 10.2.3 software.

## 3. Results

### 3.1. Design and Characterization of Influenza A-B M2e Trimer and I3-01v9a SApNPs

Our previous studies demonstrated that 24-mer ferritin (FR) and two 60-mer SApNPs, E2p and I3-01v9a, can be used to present tandem IAV M2e (human, avian/swine, and human/swine) [27]. Notably, the two 60-mer SApNPs are “multilayered” by design because they contain an inner layer of dimeric locking domains (LDs) and a hydrophobic core formed by 60 pan-DR T-helper epitopes (PADRE) [36,40,42]. These M2ex3 SApNPs conferred complete protection in mice against sequential challenges with H1N1 and H3N2 [27]. To broaden protection to cover IBVs in this study, we proposed including the IBV M2e peptide (~8 aa) in our development of a universal M2e-based vaccine. We rationally designed four influenza A-B M2e antigens comprising different arrangements of human, avian, and swine IAV M2e and three copies of human IBV M2e (i.e., M2exA3B3, M2exB3A3, M2e[AB]x3, and M2e[BA]x3), which were displayed on the I3-01v9 SApNP as vaccine candidates (Figure 1A and Appendix A). The theoretical sizes of the nanoparticles were difficult to determine due to the flexibility of six M2e tandem peptides on protein particles, but in principle, they should be larger than M2exA3 I3-01v9a-L7P (36.2 nm) reported in our previous work [27]. We also created a soluble influenza A-B M2exA3B3 trimer by fusing M2exA3B3 to a capsid-stabilizing protein of lambdoid phage 21, termed SHP (Protein Data Bank ID: 1TD0), to facilitate comparisons with various influenza A-B M2e I3-01v9 SApNPs.

Similar to our previous study [27], we systematically characterized the trimer and four SApNPs. We first transiently expressed these five constructs in 25 mL of ExpiCHO cells and purified these proteins by immunoaffinity chromatography using a mAb148 column. The elution was further analyzed by SEC (Figure 1B). The 1TD0 trimer was analyzed on a Superdex 75 Increase column, and the four SApNPs were analyzed on a Superose 6 Increase column. In the SEC profiles, the M2exA3B3-5GS-1TD0 trimer showed a higher yield than the four SApNPs, indicated by the UV_280_. The M2exA3B3-5GS-1TD0 trimer had a single SEC peak at ~8.9 mL, whereas an aggregation peak at ~8 mL and a major peak between 9.8 and 10.5 mL were observed for the four SApNPs (Figure 1B). Among the four SApNP constructs, M2e(BA)x3-5GS-I3-01v9a-L7P showed the highest yield, followed by SApNPs presenting M2exB3A3, M2exA3B3, and M2e(AB)x3 (Figure 1B). SDS-PAGE was performed to characterize all five constructs for the molecular weights (MWs) of structural monomers under reducing conditions (Appendix A). The bands showed slightly lower MWs compared to their theoretical values (22.4 kDa for M2exA3B3-5GS-1TD0 and 42.1 kDa for all SApNPs). BN-PAGE confirmed the high purity of mAb148-purified SApNPs, showing a single high-MW band with no sign of unassembled species (Figure 1C). The structural integrity of four SApNPs was validated by DLS and nsEM. DLS profiles of all four SApNPs showed similar particle sizes (61–65 nm) (Figure 1D), which is larger than influenza A M2ex3-I3-01v9a-L7P (44.8 nm) reported in our previous study [27]. The increase in hydrodynamic diameter may be due to the additional dynamic movement of M2exB3 on M2exA3 I3-01v9a-L7P SApNPs. Here, since we could not accurately calculate the theoretical sizes of the M2exA3B3 nanoparticles, we did not compare the theoretical diameters with their hydrodynamic diameters.

The nsEM images confirmed that mAb148-purified SApNP samples contained homogeneous monodispersed particles (Figure 1E), even after SApNPs were heat-treated at 70 °C for 30 min (Appendix A). Notably, the DSC and nsEM analyses suggested that SApNPs may form clusters in solution, which likely correspond to the “aggregation” peak at ~8 mL in their SEC profiles (Figure 1B). The cluster formation of influenza A M2ex3 SApNPs was also noted in our previous study [27]. The antigenicity profiles of influenza A-B M2e trimer and SApNPs were assessed at low (4 °C) and elevated (50 °C, 60 °C, and 70 °C) temperatures against mAb148 and mAb65 by ELISA (Figure 1F and Appendix A). Remarkably, the trimer and four SApNPs produced comparable EC_50_ values across the temperature range studied. The ELISA data confirmed that these influenza A-B M2e-presenting SApNPs can remain stable and antigenic at a substantially high temperature.

In summary, we successfully displayed four influenza A-B M2e antigens on our I3-01v9a SApNP platform using a design strategy similar to our previous influenza A study [27]. Extensive biochemical, biophysical, structural, and antigenic characterization indicated that all four M2e-presenting SApNPs showed high structural stability and homogeneity. Altogether, these results enabled the further evaluation of these vaccine constructs in animal models.

### 3.2. In Vivo Evaluation of Influenza A-B M2e SApNPs in Mice

The protective efficacy of influenza A-B M2e trimer and SApNPs was evaluated in BALB/c mice. We grew and propagated the mouse-adapted A/Puerto Rico/8/1934 (H1N1) and B/Florida/4/2006 (Yamagata) in MDCK cells as previously described [27]. Next, we evaluated the lethality of these viruses in mice to determine survival rates in benchmark challenge studies (Figure 2A). We then used the Reed–Muench and Spearman–Karber methods [51,52] to calculate the viral dilutions to produce an LD_50_ in mice. We determined the LD_50_ to be 4217× and 5.2× dilutions for A/Puerto Rico/8/1934 (H1N1) and B/Florida/4/2006, respectively. Based on these LD_50_ values, we determined the appropriate LD_50_ × 10 and LD_50_ × 5 challenge doses for evaluating the protective efficacy of the influenza A-B M2e immunogens.

To determine the protective efficacy of the M2exA3B3-5GS-1TD0 trimer and four SApNP vaccines, we formulated the antigens with AddaVax adjuvant (oil-in-water emulsion) and immunized the mice intradermally through the footpads (4 footpads, 2.5 μg/footpad) at weeks 0 and 3, followed by the first virus challenge with IAV or IBV at week 6 and the second virus challenge with IBV or IAV in the surviving mice, respectively (Figure 2B). Survival rates and weight loss were measured daily for 14 days after each viral challenge (Figure 2C). After the first challenge with A/Puerto Rico/8/1934 (H1N1), all naïve mice (negative control) succumbed by day 8, whereas 100% of the positive control mice, which were immunized with a single dose of strain-matched inactivated A/Puerto Rico/8/1934 (H1N1) virus, survived the challenge. Among the M2e-immunized groups, 10% of the mice survived in the M2exA3B3 trimer group. In contrast, groups that were vaccinated with various influenza A-B M2e SApNPs showed significantly higher survival rates (70–100%). Notably, the protection appeared to correlate with the sequence arrangement of IAV and IBV M2e presented on the I3-01v9a SApNP surface. The M2exA3B3 SApNP group showed the strongest protection (100%), whereas the M2e(BA)x3 group showed the lowest protection (70%) among all SApNP constructs. We observed a similar trend in the average peak weight loss among all groups. The naïve mouse group showed the highest weight loss (27.9 ± 1.7%), whereas the positive control mice lost the least weight (9.9 ± 5.8%). Among the AddaVax-adjuvanted M2e immunogen groups, the M2exA3B3 trimer group lost 26.6 ± 2.6% of their body weight, whereas the M2exA3B3 SApNP group showed the least weight loss (18.0 ± 5.3%). The other three SApNP groups showed slightly higher weight loss than the M2exA3B3 SApNP group, with the average total weight loss ranging from 21.4% to 22.4%.

In addition to a new naïve group, the surviving mice from the A/Puerto Rico/8/1934 (H1N1) challenge were then challenged with B/Florida/4/2006 (Yamagata) (Figure 2C). All mice in the naïve group succumbed to the infection 8 days post-challenge with B/Florida/4/2006 (Yamagata). Among the immunized mice, the lowest survival rate was obtained for the inactivated A/Puerto Rico/8/1934 (H1N1) group, with only 20% of mice surviving a heterotypic challenge. Among the influenza A-B M2e groups, all mice that survived the previous influenza A challenge survived the B/Florida/4/2006 (Yamagata) challenge. The weight loss observed following the influenza B challenge showed a similar trend to the survival rate, with the naïve group showing the most weight loss (27.0 ± 1.0%) followed by the inactivated A/Puerto Rico/8/1934 (H1N1) group (25.8 ± 3.2%). In contrast, weight loss in the M2e groups ranged from 5.0 ± 2.7% in the M2exB3A3 SApNP group to 14.5% in the M2exA3B3 trimer group.

Next, we evaluated the protective efficacy of M2e groups against sequential challenges with B/Florida/4/2006 (Yamagata) followed by A/Puerto Rico/8/1934 (H1N1) (Figure 2D). As expected, 100% of the mice immunized with a single dose of inactivated B/Florida/4/2006 (Yamagata) survived the strain-matched challenge. In mice vaccinated with AddaVax-adjuvanted influenza A-B M2e trimer and I3-01v9a SApNP constructs, 60% of the M2exA3B3 trimer group survived the challenge. Among the SApNP groups, a slightly higher survival rate of 70% was observed for all four M2e I3-01v9a SApNP groups. In terms of peak weight loss, the mice vaccinated with the strain-matched inactivated B/Florida/4/2006 (Yamagata) lost the least body weight (6.0 ± 4.8%). All M2e trimer and SApNP groups lost a greater percentage of their body weight (19.4–21.9%). We next conducted a second challenge in surviving mice with a lethal dose of A/Puerto Rico/8/1934 (H1N1) (Figure 2D). All inactivated B/Florida/4/2006 (Yamagata)-immunized mice succumbed to the challenge. In contrast, 60% of mice in the M2exA3B3 trimer group that survived the previous IBV challenge survived the A/Puerto Rico/8/1934 (H1N1) challenge. Among the influenza A-B M2e SApNP groups, only the M2exA3B3 SApNP group showed complete protection (100%), whereas a survival rate of 85.7% was observed in the three remaining SApNP groups. Weight loss showed a similar trend. The inactivated B/Florida/4/2006 (Yamagata) group and naïve group showed the highest level of weight loss (26.5–27.9%), followed by the M2exA3B3 trimer group (19.6 ± 6.9%). Among the influenza A-B M2e-presenting I3-01v9a SApNP groups, the M2exA3B3 group showed relatively lower weight loss (12.7 ± 4.3%).

In summary, the M2exA3B3-presenting I3-01v9a SApNP formulated with AddaVax adjuvant significantly outperformed the soluble M2exA3B3 trimer upon prime-boost vaccination, showing a higher survival rate and lower weight loss in the sequential A/Puerto Rico/8/1934 (H1N1) and B/Florida/4/2006 (Yamagata) challenge studies. Among the four SApNP groups based on different sequence arrangements of IAV and IBV M2e, the M2exA3B3 SApNP group showed superior (100%) protection against A/Puerto Rico/8/1934 (H1N1) challenge compared to the other three designs, and all four constructs showed similar protection (70%) against B/Florida/4/2006 (Yamagata) challenge. In contrast, the inactivated vaccines only protected against the strain-matched virus.

### 3.3. Evaluation of Influenza A-B M2e Vaccine-Induced Antibody Responses

We assessed the M2e binding antibody response in mouse serum using the sequence-matched M2e trimer and determined EC_50_ titers by ELISA (Figure 3A,B and Appendix A). For example, the M2exA3B3-5GS-foldon trimer was used as a coating antigen for the M2exA3B3-5GS-1TD0 trimer and M2exA3B3 I3-01v9a SApNP groups, and the M2e(AB)x3-5GS-foldon trimer was used to analyze the M2e(AB)x3 I3-01v9a SApNP group. M2exA3B3 presented on the I3-01v9a SApNP elicited superior binding antibody responses in immunized mice at both weeks 2 and 5, showing 11.3- and 10.1-fold higher EC_50_ titers, respectively, compared to the corresponding 1TD0 trimer (*p* < 0.0001). Although the coating antigens were different, the EC_50_ titers of the four I3-01v9a SApNP groups against the sequence-matched M2e trimer probes showed significantly higher values than the M2exA3B3 1TD0 trimer group at all time points. Among the four SApNPs, the M2exA3B3 I3-01v9a SApNP yielded the highest EC_50_ titer at week 5.

The recognition of homotetrameric M2e, which represents the “native” M2e conformation during influenza infection, by M2e-immune sera was assessed for the AddaVax-adjuvanted M2eA3B3 1TD0 trimer and I3-01v9a SApNP groups (Figure 3C). In this experiment, ELISA was performed to assess immune-sera binding to M2e presented on MDCK cells infected by IAVs A/Puerto Rico/8/1934 (H1N1) and A/Hong Kong/1/1968 (H3N2) as well as IBVs B/Florida/4/2006 (Yamagata lineage) and B/Brisbane/60/2008 (Victoria lineage). The absorbance (at 450 nm) of binding to M2e expressed on both IAV- and IBV-infected MDCK cells was up to two-fold higher for the M2eA3B3 I3-01v9a SApNP group compared to M2exA3B3 1TD0 trimer at the lowest serum dilution tested. An influenza A hM2e antibody in the immunoglobulin form, mAb148, showed binding to IAVs but no detectable binding to IBVs, confirming that this antibody is IAV M2e-specific. There is currently no reported M2e-specific antibody targeting influenza B. These results demonstrated that the M2exA3B3 timer and SApNP immunogens elicited antibodies against both native IAV M2e and IBV M2e, which have minimal antigenic overlap as they have distinct amino acid sequences. A naïve mouse serum sample was included as a negative control, which showed negligible binding to all four viruses tested in this experiment. Overall, the M2exA3B3-presenting I3-01v9a SApNP vaccine appeared to be more immunogenic than its respective trimer form in vivo.

## 4. Discussion

Influenza is an acute respiratory illness that infects 1 billion people each year, with 3–5 million cases of severe disease and upwards of 600,000 deaths globally [6]. Approximately 75% of infections are attributed to IAVs and 25% are attributed to IBVs [8,9]. IAV is also responsible for periodic pandemics caused by novel influenza strains arising from the reassortment of IAV genomes from different host species [53,54]. Inactivated influenza vaccines have only 10–60% efficacy against seasonal influenza and do not protect against novel pandemic strains. It is therefore imperative to develop a universal influenza vaccine capable of protecting against diverse seasonal and potential pandemic strains [12,13]. M2e presents an attractive target for the development of such universal vaccines because of its high sequence conservation within IAVs, as well as IBVs. Although M2e is small (~23 aa for IAV and ~8 aa for IBV) and poorly immunogenic in the context of natural infection, it can be attached to large protein scaffolds and non-protein carriers to elicit cross-protective non-neutralizing antibodies that significantly reduce viral replication and disease severity, as previously shown with IAV M2e vaccines [14,15,16,17,18,19].

We recently developed a tandem M2ex3 antigen combining IAV M2e from human, avian, and swine hosts [27]. This M2e antigen was genetically fused to the I3-01v9a SApNP backbone and expressed in CHO cells to produce homogenous nanoparticles displaying 60 copies of M2ex3. Formulated with a conventional squalene-based adjuvant (AddaVax), M2ex3 I3-01v9a SApNPs were retained in lymph node follicles for 8 weeks or longer and generated strong germinal center reactions, M2e-specific antibody responses, antibody-dependent cellular cytotoxicity, and functional CD4^+^ and CD8^+^ T cell responses, resulting in 100% protection of immunized mice from sequential lethal challenges with H1N1 and H3N2, the IAV subtypes currently circulating in humans [27]. To expand the scope of protection, we incorporated IBV M2e in a series of four influenza A-B M2e constructs comprising different arrangements of three IAV M2e domains plus three copies of IBV M2e and displayed them on an I3-01v9a SApNP. We demonstrated that tandem repeats of influenza A and B M2e epitopes attached to a large multilayered 60-mer SApNP were more protective than a soluble M2e trimer. Notably, the M2exA3B3 I3-01v9 SApNP adjuvanted with AddaVax elicited 100% protection against a lethal IAV (H1N1) challenge and 70% protection against a lethal IBV (Yamagata lineage) challenge in mice, providing proof-of-concept for an M2e-based universal influenza vaccine targeting both IAV and IBV. In contrast, the inactivated vaccines tested were strain- and type-specific, showing minimal protection against a non-strain-matched challenge. From an antigenicity perspective, the sequence arrangement of IAV and IBV M2e on the SApNP platform likely plays an important role in M2e-mediated protection. Only mice immunized with M2exA3B3 SApNPs, which present three IAV M2e domains outermost from the particle surface (at the N-terminus of the antigen), were 100% protected against the initial challenge with H1N1, whereas the other three SApNP groups with various IAV and IBV M2e arrangements showed 70–80% protection. Notably, mice vaccinated with M2exA3B3 SApNPs were also fully protected against a second challenge with either IAV (H1N1) or IBV (Yamagata lineage). However, the location of IBV M2e does not appear to influence protective efficacy against IBV as all four SApNP designs showed 70% protection after the first challenge with IBV. The short length of IBV M2e, which is less than one-third that of IAV M2e, may explain why the maximum protective efficacy achieved against the IBV challenge was only 70%. Importantly, this degree of protection against influenza B using IBV M2e has not been previously demonstrated and presents a first step toward a truly universal M2e-based vaccine against both influenza A and B. In summary, the immune recognition of IAV and IBV M2e epitopes may be impacted by the length, sequence, and position of these M2e peptides when they are displayed on the surface of a particle. These factors must be carefully evaluated in the development of future influenza A-B M2e vaccines.

Several aspects of this study can be explored further in the future. First, incorporating more copies of IBV M2e in SApNPs may improve the protective efficacy they elicit in immunized animals against a lethal IBV challenge. Second, the vaccine constructs in the present study were expressed in CHO cells. Alternative cost-effective and scalable expression systems, such as *E. coli* and yeast, can be explored for producing M2e-based SApNP vaccines. Third, IAV and IBV M2e sequences can be incorporated in HA or HA stem-based immunogens to induce potent and broadly protective antibody responses to block viral entry and reduce viral replication and disease severity [55,56]. Furthermore, studies to elucidate the interactions between A-B M2e SApNPs and cellular components in the lymph nodes, like those we previously conducted for influenza A M2e SApNPs [27], are warranted. Fourth, an optimized influenza A-B M2e SApNP construct may be used alone or as a component of a multivalent seasonal vaccine to boost the breadth of protection against circulating and pandemic influenza strains. This would be especially advantageous for reducing influenza cases in years when the seasonal vaccine is poorly matched to circulating strains or if an influenza pandemic should arise.

## 5. Conclusions

The influenza A-B M2e-based SApNP vaccines developed in this study were shown to elicit robust M2e-specific immunity and protection against lethal influenza A and B challenges in a mouse model. For the first time, we demonstrate how IBV M2e can be utilized as part of a vaccine antigen to protect against a lethal influenza B challenge. These well-characterized influenza A-B M2e SApNP immunogens represent the first step toward a truly universal influenza vaccine targeting the two major types of influenza that circulate in humans.

## Figures and Tables

**Figure 1 vaccines-12-00975-f001:**
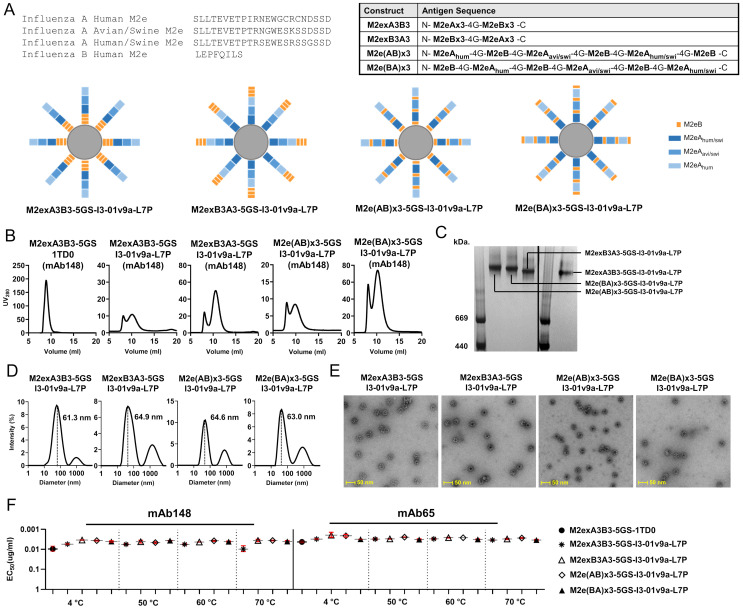
Design and in vitro characterization of influenza A-B M2e-based immunogens. (**A**) **Top left**: Amino acid sequences for influenza A M2e from human (UniProt: P06821, residue 2–24), avian/swine (residue 2–24) [16], and human/swine (residue 2–24) [16] and influenza B M2e from human (UniProt: A4D452, residue 2–9). **Top right**: Series of designed antigen sequences for influenza A-B M2exA3B3, M2exB3A3, M2e(AB)x3, and M2e(BA)x3 constructs. A glycine (G4) linker was added between any two adjacent M2e peptides, regardless of IAV and IBV M2e. **Bottom**: Schematics of influenza A-B M2e antigens consisting of various arrangements of human, avian, and swine IAV M2e plus three copies of IBV M2e displaying on I3-01v9a-LD7-PADRE (or simply I3-01v9a-L7P) SApNPs. (**B**) SEC profiles of the M2exA3B3-5GS-1TD0 trimer and M2exA3B3-, M2exB3A3-, M2e(AB)x3-, and M2e(BA)x3-presenting I3-01v9a SApNPs. The trimer was analyzed on a Superdex 75 Increase 10/300 GL column and four SApNPs were analyzed using a Superose 6 Increase 10/300 GL column. (**C**) BN-PAGE for mAb148-purified influenza A-B M2e I3-01v9a SApNPs. (**D**) DLS profiles of mAb148-purified influenza A-B M2e I3-01v9a SApNPs. Average particle sizes derived from DLS are labeled. (**E**) nsEM micrographs of mAb148-purified influenza A-B M2e I3-01v9a-L7P SApNPs. (**F**) ELISA analysis of the M2eA3B3-5GS-1TD0 trimer and M2exA3B3-, M2exB3A3-, M2e(AB)x3-, and M2e(BA)x3-presenting I3-01v9a SApNPs binding to mAb148 (**left**) and mAb65 (**right**) before and after heating to 50 °C, 60 °C, and 70 °C for 30 min.

**Figure 2 vaccines-12-00975-f002:**
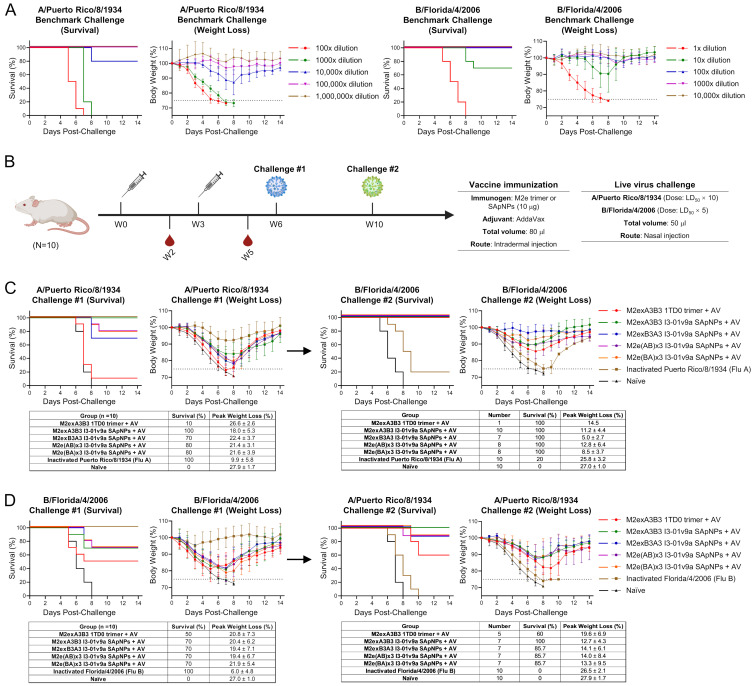
Protective efficacy of influenza A-B M2e-based trimer and SApNP vaccines in mice. (**A**) Determination of lethal dose 50 (LD_50_) for mouse-adapted A/Puerto Rico/8/1934 (H1N1) and mouse-adapted B/Florida/4/2006 (Yamagata lineage) through benchmark challenge studies (*n* = 10 mice/virus dilution). Mice were challenged with various dilutions of virus stock through nasal injection, with survival, weight loss, and morbidity monitored for 14 days post-challenge. (**B**) Schematic representation of mouse immunization, blood collection, and sequential challenges. Mice were immunized with the influenza A-B M2e 1TD0 trimer or SApNPs adjuvanted with AddaVax (AV) and then challenged using the LD_50_ × 10 of mouse-adapted A/Puerto Rico/8/1934 (H1N1) or LD_50_ × 5 of mouse-adapted B/Florida/4/2006 (Yamagata lineage) at weeks 6 or 10. (**C**) Survival rates and weight loss of vaccinated mice after the first challenge with mouse-adapted A/Puerto Rico/8/1934 (H1N1) followed by the second challenge with mouse-adapted B/Florida/4/2006 (Yamagata lineage). Mice were monitored for survival, weight loss, and morbidity for 14 days. Tested groups (*n* = 10 mice/group) included mice immunized with influenza A-B M2exA3B3 1TD0 trimer and four SApNPs (10 μg/mouse), naïve mice as a negative control, and strain-matched inactivated A/Puerto Rico/8/1934 virus (10 μg/mouse) as a positive control for the first mouse-adapted A/Puerto Rico/8/1934 (H1N1) challenge. (**D**) Survival rates and weight loss of vaccinated mice after the first challenge with mouse-adapted B/Florida/4/2006 followed by the second challenge with mouse-adapted A/Puerto Rico/8/1934 (H1N1). A group in which mice were immunized with inactivated B/Florida/4/2006 (Yamagata) virus was included as a positive control for the first challenge with strain-matched mouse-adapted B/Florida/4/2006. Images of mouse immunization, virus challenge, and blood collection were created with BioRender.com (accessed on 21 May 2024).

**Figure 3 vaccines-12-00975-f003:**
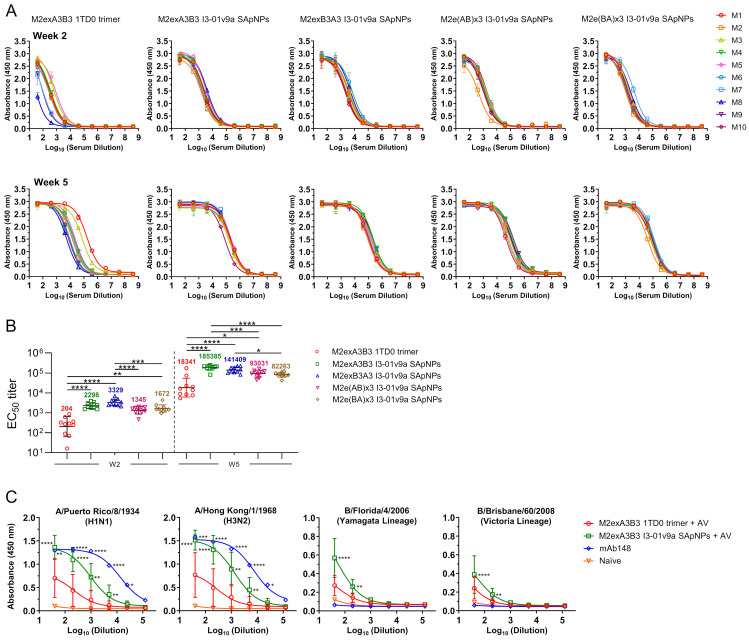
Immunogenicity of influenza A-B M2e-based trimer and SApNP vaccines assessed by serum binding to M2e trimer probes and homotetrameric M2e on influenza-infected cells. (**A**) ELISA curves showing the influenza A-B M2e-immune sera of individual mice (*n* = 10 mice) binding to sequence-matched M2e coating antigen (M2exA3B3-5GS-foldon, M2exB3A3-5GS-foldon, M2e[AB]x3-5GS-foldon, and M2e[BA]x3-5GS-foldon). (**B**) Influenza A-B M2e vaccine-induced serum binding antibody responses at weeks 2 and 5 were measured by half-maximal effective concentration (EC_50_) titers. The assay was performed in duplicate with a starting serum dilution of 40× followed by seven 10-fold titrations. (**C**) Serum binding to M2e on the surface of MDCK cells infected with two IAVs, A/Puerto Rico/8/1934 (H1N1) and A/Hong Kong/1/1968 (H3N2), and two IBVs, B/Florida/4/2006 (Yamagata lineage) and B/Brisbane/60/2008 (Victoria lineage). Antibody mAb148 was used as a positive control for IAVs and a negative control for IBVs. The assay was performed in duplicate with a starting serum dilution of 40 times followed by five 5-fold titrations. Antibody mAb148 was diluted to 10 μg/mL followed by five 5-fold titrations. The statistical analysis was performed using one-way ANOVA followed by Tukey’s multiple-comparison *post hoc* test for each time point (**B**) and influenza virus strain (**C**). For significance, ns (not significant), * *p* < 0.05, ** *p* < 0.01, *** *p* < 0.001, and **** *p* < 0.0001.

## Data Availability

All data to understand and assess the conclusions of this research are available in the main text and in the Appendix A.

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
