# Peer review of "A Single-Component Multilayered Self-Assembling Protein Nanoparticle Vaccine Based on Extracellular Domains of Matrix Protein 2 against Both Influenza A and B"

_vaccines, 2024, doi:10.3390/vaccines12090975_

Round 1

Reviewer 1 Report

Comments and Suggestions for Authors

Dear Authors,

The authors present a novel research article discussing the development and efficacy evaluation of a universal, single-component multilayered self-assembling protein nanoparticle vaccine based on extracellular domains of matrix protein 2 against both influenza A and B. The work compares the therapeutic effect of a series of antigens incorporated different arrangements of three IAV M2e domains and three copies of IBV M2e against Influenza A and B. The authors cover an important topic in current research and developed a universal influenza vaccine.

Antigens were expressed in CHO cells and purified from media by an antibody based binding column. Purity and quantity was assessed by gel electrophoresis, dynamic light scattering, electron microsccopy and ELISA. Infectious and inactivated influenza virus were prepared for further assays. Using the standard methods, mice were immunized, serum collected for antibody quantification. Against LD50 influenza infected mice, the A-B M2e-based trimer and SApNP mixed with adjuvant reduced mortality. In vitro, the vaccines induced serum binding response on the surface of influenza infected MDCK cells.

The techniques employed are appropriate and appear to be performed to a very high standard. The text is easy to read and comprehensive. Figures and diagrams are all excellent.

Comments

1. How are the vaccine stored after construction? Please clarify in the method section.

2. In the Methods, the authors described clearly on constructing the  and testing in vivo efficacy. It would be useful to provide catalogue numbers of the important reagents, so the researchers who interested can reproduce the vaccine.

3. In the result section, when the authors mention the comparisons between different groups, the authors used “higher”, “highest” to describe the difference between groups, however this is not very clear to readers if that is statistical significant. Can the authors clarify if the comparison is statistical significant and include the significance level (eg. p<0.01 or no significant difference etc.) in the text.

4. Line 36, 154, 282-289 etc. there seems to be some font changes, possibly due to copy and paste.

5. Line 509-510, please add reference for this statement.

Regards

Reviewer 2 Report

Comments and Suggestions for Authors

The authors develop and evaluated a universal influenza vaccine based on single-component self-assembling protein nanoparticles (1c-SApNPs). The authors demonstrated that the newly develop can elicit robust M2e-specific immunity and protection against lethal influenza A and B challenges in a mouse model. This study provides new insights to the development of influenza vaccines. Please see suggestions below.

The conclusion is that the vaccine is a universal vaccine against both influenza A and B, which is not fully supported by the evidence collected in the study. the mice study only used PR8 and B/Florida/4/2006, other mice-adapted viruses are required to confirm the findings. These results are the critical information to establish their conclusion. Without the additional experiments, it would be safe to avoid stating “universal vaccine against influenza A and B” in this manuscript. In the abstract, line 27-28,  please replace “H1N1”  and “Yamagata lineage” by strain names.

Reviewer 3 Report

Comments and Suggestions for Authors

The study of the vaccine potential of multilayered self-assembling protein nanoparticle is of undoubted interest to readers and was carried out at a high scientific level. This study is worthy of publication in the journal "Vaccines".

I have a few questions for the authors that do not affect the overall high assessment of their work.

Figure 1 shows the theoretical diameters of the nanoparticles. How were these parameters calculated? Why do theoretical data differ from hydrodynamic radii by almost two times? (It is unlikely that hydration shells provide an increase of almost 30 nm).

Lines 337-339: “The bands showed slightly lower molecular weights (MWs) compared to their theoretical values ​​(22.4 kDa for M2exA3B3-5GS-1TD0 and 42.1 kDa for all SApNPs)” - apparently, this means the molecular weight of structural monomers, and not the particles themselves. Please clarify this.

Reviewer 4 Report

Comments and Suggestions for Authors

In this manuscript, based on a single-component self-assembling protein nanoparticle (1c-SApNP) platform, Zhang and colleagues constructed four SApNPs presenting M2e ectodomains of IAV and IBV in different arrangements. They evaluated the immunogenicity of SApNPs in mice and protective efficacy in mice against IAV and IBV. Finally, they found the M2exA3B3 SApNP was the most effective vaccine candidate for IAV and IBV. This work provided insights into developing broad-spectrum vaccines against IAV and IBV. After reviewing this manuscript, I have some minor suggestions as follows:

1. In this manuscript, most methods and results are described in detail. But I think descriptions of some of the experimental protocols and results are not necessary, such as detection of LD50 of virus.

2. Lines 234 and 235, “spun down,” mean “centrifuge”?

3. Line 283 to 289, the font and size of the text appear to be inconsistent.
